# The Intelligent Path Planning System of Agricultural Robot via Reinforcement Learning

**DOI:** 10.3390/s22124316

**Published:** 2022-06-07

**Authors:** Jiachen Yang, Jingfei Ni, Yang Li, Jiabao Wen, Desheng Chen

**Affiliations:** 1School of Electrical and Information Engineering, Tianjin University, Tianjin 300072, China; yangjiachen@tju.edu.cn (J.Y.); nijingfei@tju.edu.cn (J.N.); wen_Jiabao@tju.edu.cn (J.W.); chendesheng@tju.edu.cn (D.C.); 2College of Mechanical and Electrical Engineering, Shihezi University, Shihezi 832003, China

**Keywords:** reinforcement learning, agricultural robot, path planning, obstacle avoidance, intelligent control, Internet of Things

## Abstract

Agricultural robots are one of the important means to promote agricultural modernization and improve agricultural efficiency. With the development of artificial intelligence technology and the maturity of Internet of Things (IoT) technology, people put forward higher requirements for the intelligence of robots. Agricultural robots must have intelligent control functions in agricultural scenarios and be able to autonomously decide paths to complete agricultural tasks. In response to this requirement, this paper proposes a Residual-like Soft Actor Critic (R-SAC) algorithm for agricultural scenarios to realize safe obstacle avoidance and intelligent path planning of robots. In addition, in order to alleviate the time-consuming problem of exploration process of reinforcement learning, this paper proposes an offline expert experience pre-training method, which improves the training efficiency of reinforcement learning. Moreover, this paper optimizes the reward mechanism of the algorithm by using multi-step TD-error, which solves the probable dilemma during training. Experiments verify that our proposed method has stable performance in both static and dynamic obstacle environments, and is superior to other reinforcement learning algorithms. It is a stable and efficient path planning method and has visible application potential in agricultural robots.

## 1. Introduction

With the development of artificial intelligence technology and the increasing material and spiritual needs of humans, Internet of Things (IoT) technology is maturing [1]. Increasing modernization and intelligence in all walks of life is a trend that conforms to the development of the times. In addition, agriculture is the top priority of all industries and the most indispensable demand of humans. Determining how to improve agricultural efficiency and increase crop yield is one of the most urgent issues in the current era. In recent years, artificial intelligence technology has brought new solutions to this problem. Promoting agricultural modernization, saving human labor, and improving agricultural efficiency are the advantages of artificial intelligence.

At present, many scholars have carried out intelligence research [2] and IoT research [3] in agricultural systems. In the work of [4], they study the perception system of crop nutrient elements for agricultural robots, which lays the foundation for agriculturally precise fertilization and saves human labor. In [5], they build a test platform for agricultural mobile robots, using multi-sensor fusion and a self-encoding network to predict the robot’s high-precision position, and prevent the location accuracy of the robot being affected by signal interference. RGB-D cameras are used to process the scene image to perceive and detect the obstacles in the image, so that the automatic lawnmower can cope with the challenging work area [6]. In the study by [7], they review the IoT systems in agriculture, and analyze the status and challenges of automatic harvesting robots and automatic picking robots. Additionally, ref [8] uses CNN and SSD model to detect the life cycle of tomatoes in greenhouses and improves the perception ability of picking robots. The above research reflects the necessity and effectiveness of artificial intelligence technology in the agricultural field, and most task-oriented agricultural robots are inseparable from the problem of autonomous path planning. Therefore, this paper focuses on the path planning of agricultural robots, and uses reinforcement learning methods to train agricultural robots to plan paths autonomously, so as to complete a series of agricultural tasks.

Traditional path planning methods such as potential field method [9] and A* search method [10] are mature and widely-used methods that have been used for quite some time. Refs. [11,12] carried out dynamic planning for system scheduling and reasonably handled route states to avoid deadlock, which is a classic planning problem. Lin et al. took the UAV as the research object, and adopted the RRT algorithm to complete offline path planning. During the navigation process, they used the vision-based obstacle detection technology to achieve online obstacle avoidance [13]. Han et al. combined the ant colony algorithm for full coverage path planning and obstacle avoidance for the underwater glider in the ocean area [14]. These methods perform well in their applicable scenarios, but there are still some limitations. When the target point or obstacles change slightly, these methods need to re-plan because of the lack of adaptability and flexibility. With the rapid development of deep learning in recent years, intelligent methods have played an increasingly important role in smart agriculture [15,16], image screening [17], environmental monitoring [18], edge computing [19], and path planning [20]. At the same time, it provides an effective technical means for agricultural production, automatic driving, and other fields. Reinforcement learning is a method that is suitable for the environments for which there is a lack of prior knowledge, and learns through the interaction with the environment. Therefore, in order to make the robot have certain flexibility and autonomy, more and more scholars use deep reinforcement learning to train the robot, which has the ability to plan autonomously and tolerate some subtle changes. As a result, it has higher efficiency and better adaptability than traditional methods.

Bianca et al. adopt a dual-mode combination of classical planning algorithms and deep reinforcement learning methods to realize the robot’s obstacle avoidance, and achieve the mechanism of offline training and real-time execution [21]. David et al. combine unsupervised learning with reinforcement learning to achieve robust multi-agent path planning in a two-dimensional unknown environment [22]. The globally guided reinforcement learning method also has good versatility. Wang et al. make full use of the environmental spatiotemporal information to solve the robot path planning problem in various scenarios [23]. Yao et al. combine the black-hole potential field with reinforcement learning to solve the problem that the agent can easily fall into the local stable point, and adopt curriculum learning to learn a series of obstacle avoidance tasks from easy tasks to difficult tasks [24]. Yan et al. use the D3QN method in reinforcement learning to study the evasion path planning of UAV when encountering enemy detection and attack, and simulate the UAV’s survival route based on a simulation platform [25]. Guo et al. [26] and Chen et al. [27], respectively, use a DDPG algorithm and a Q-learning algorithm in reinforcement learning to study the autonomous navigation of ships. They ensure that ships follow navigation rules and drive safely. Most of the research objects of path planning are UAVs, unmanned ships, or miniature ground robots, and there are few studies on agricultural scenarios. Aiming at the lack of research on the path planning problem of agricultural robots, this paper adopts a reinforcement learning method to carry out a series of studies on obstacle avoidance and path planning tasks.

The main contributions of this paper are as follows.

Currently, reinforcement learning algorithms used in path planning are mainly earlier algorithms. This paper proposes a novel Residual-like Soft Actor Critic (R-SAC) algorithm, which combines the residual-like structure with the SAC algorithm to optimize the network structure of SAC. The improved algorithm is able to complete dynamic obstacle avoidance tasks. Experiments show that our method is more efficient than other reinforcement learning algorithms.Aiming at the time-consuming problem of the interaction process of reinforcement learning, a supervised learning method based on offline expert experience is proposed as a pre-training of reinforcement learning. This method greatly reduces the training time and improves the efficiency of learning, and realizes an efficient mechanism of single exploration and multiple utilization.The one-step TD error of the SAC algorithm only considers the reward of the current step and ignores the periodic or long-term reward. As a result, if the agent misses the target point, it may not adjust in time, which reduces the success rate of the task. Therefore, we add a multi-step TD error mechanism, so that the agent can comprehensively consider the long-term reward, and can choose the optimal adjustment policy in the inferior situation.

The rest of this paper is arranged as follows. Section 2 introduces the related work of path planning in the agricultural field and the current progress of reinforcement learning in path planning. Section 3 introduces the R-SAC algorithm, offline expert experience method and multi-step TD error proposed in this paper. In Section 4, we conduct the static obstacle avoidance experiments and mobile obstacle avoidance experiments. We also compare the proposed algorithm with other reinforcement learning methods. In addition, we conduct multi-step contrast experiments and ablation experiments of R-SAC algorithm to discuss the improved mechanism in Section 5. Lastly, we summarize the full work in Section 6.

## 2. Related Work

Regarding path planning problems, reinforcement learning methods have been verified and recognized by many scholars, but in terms of the algorithm, most of them are still at the primary level. Hyansu et al. apply DQN to complete the multi-agent path planning task, and use convolutional neural networks to process image input [28]. Lei et al. utilize the DDQN algorithm to enable the agent to achieve local path planning in an unknown dynamic environment, and combine it with the ROS framework for physical verification [29]. In the study of Beomjoon et al., they utilize a feature extraction module to extract local features, and then use inverse reinforcement learning to learn a cost function from human experience to assist local path planning, and make the agent trajectory similar to human behavior to improve the safety of the robot [30]. Zhou et al. combine DDPG with ANFIS network to realize real-time dynamic planning of vehicles to maximize vehicle energy efficiency and state of charge [31]. Lin et al. combine DDPG with LSTM network to fully memorize and utilize the robot’s past state in order to complete the collision-free path planning and picking of tasks of picking robots in the orchard [32]. Although these earlier reinforcement learning algorithms can achieve tasks, there are still many problems that can be optimized, and the update and innovation of algorithms can help robots complete more difficult tasks.

In the agricultural field, Santos et al. make a summary review of the current agricultural path planning applications. They analyze and compare a variety of point-to-point methods and coverage path planning methods, and conclude that path planning applications in the agricultural field are very few, and the methods need to be optimized [33]. Han et al. adopt a path tracking method based on slip estimation to implement a path planner for agricultural robots with auto-steering control, which reduces the restriction of the shape of the farmland [34]. Saba et al. apply a model-based reinforcement learning method to build an accurate environment model in an unknown dynamic environment to enable multiple UGVs to learn an environment map, and then implement path planning in the established environment model [35]. In [36], the 3D farmland terrain required by agricultural robots or autonomous tractors is considered by Hameed et al., and a 3D coverage method is proposed to avoid skips or overlaps in 3D terrain. Wu et al. use a 3D scanner to collect terrain and obstacle information and establish a spatial model of hill areas, and then adopt an ant colony algorithm to complete path planning in 3D space [37]. In the study of [38], Han et al. model the dynamics of agricultural robots, and develop a method that can control the angle and speed of the robot. Liu et al. adopt the fuzzy clustering method to realize the obstacle avoidance of the robot, and fuse multiple sensors to collect information to assist obstacle avoidance, which performs well in physical experiments [39]. Li et al. apply a DQN algorithm to complete the intelligent path tracking task based on the agricultural machinery driving scenario, and maintain a fast convergence speed and good stability in the dynamic environment [40].

The above studies demonstrate the feasibility of various methods in path planning. Most of traditional methods require pre-planning, and require knowledge of accurate map information before planning. When some factors change in the environment, such as movable obstacles, these methods cannot adjust in time and even need to re-plan as a new environment. For reinforcement learning, due to its characteristics of real-time interaction and learning from environmental feedback, changes in the environment can be acquired and the policy can be adjusted in real time. Therefore, reinforcement learning has a certain tolerance for dynamic changes or disturbances in the environment, and has obvious advantages and adaptability in dynamic environments. Compared with the existing application of reinforcement learning in the agricultural field, the method in this paper speeds up the algorithm convergence. It improves the performance in dynamic obstacle avoidance scenarios, and can be applied to more complex path planning tasks.

## 3. Algorithm

The principle of reinforcement learning is that the agent interacts with the environment in real time and learns to make decisions. It models and solves problems based on the Markov Decision Process (MDP). $\left(S,A,P,R,\gamma \right)$ are key elements in the interaction process of reinforcement learning, where $S=\left\{s_{1},s_{2},\cdots ,s_{t}\right\}$ represents the state space of the environment, including the state information of the agent at each step. $A=\left\{a_{1},a_{2},\cdots ,a_{t}\right\}$ represents the action space of the agent, including the range of the agent’s action choices. *P* is the state transition probability, and $R=\left\{r_{1},r_{2},\cdots ,r_{t}\right\}$ is the feedback of the environment based on the state and chosen action at each step, from which the agent learns the policy. γ is the discount factor for discount calculation of the cumulative rewards, and the cumulative rewards can be expressed as
(1)$$\begin{array}{cc} R^{\left(t\right)} & =r^{\left(t+1\right)}+\gamma r^{\left(t+2\right)}+\gamma^{2}r^{\left(t+3\right)}+\cdots \\ \; & =\sum_{n=0}^{\infty }\gamma^{n}r^{\left(t+n+1\right)} \end{array}$$

The basic framework of reinforcement learning is shown in Figure 1.

### 3.1. R-Sac Algorithm

In this paper, the R-SAC algorithm is proposed for the path planning of the agent. The SAC algorithm is based on the actor-critic framework, where the actor network outputs the policy π and is updated by the policy gradient method. The parametric representation of policy π is expressed as
(2)$$\pi_{\theta }\left(a_{t}|s_{t}\right)=P_{\theta }\left\{A=a_{t}|S=s_{t}\right\}$$
where θ is the parameter vector of the policy π, and $P_{\theta }$ is the probability of output action $a_{t}$ based on state $s_{t}$. The state-action pairs in a complete episode constitute the trajectory τ, and its probability is expressed as
(3)$$\begin{array}{cc} \pi_{\theta }\left(\tau \right) & =p_{\theta }\left(s_{1},a_{1},\cdots ,s_{n},a_{n}\right)\\ \; & =p_{\theta }\left(s_{1}\right)\prod_{t=1}^{n}\pi_{\theta }\left(a_{t}|s_{t}\right)p_{\theta }\left(s_{t+1}|s_{t},a_{t}\right) \end{array}$$

The essential optimization goal of reinforcement learning is to maximize the cumulative reward
(4)$$J\left(\theta \right)=E\left[R\left(\tau \right)\right]$$

The optimization idea of the policy gradient method is to find a set of optimal parameters θ in the parameterized policy function to maximize the cumulative reward.
(5)$$\theta^{*}=\underset{\theta }{\arg\max}E_{\tau }\left[\sum_{t=1}^{n}R\left(s_{t},a_{t}\right)\right]$$

Therefore, the gradient of the optimization goal $J\left(\theta \right)$ is the policy gradient, and the policy gradient is approximated by the mean value of *N* samples:(6)$$\begin{array}{cc} \nabla_{\theta }J\left(\theta \right) & =\frac{1}{N}\sum_{N}\left[\sum_{t=1}^{n}\nabla_{\theta }\log\pi_{\theta }\left(a_{t}|s_{t}\right)\right]\left[\sum_{t=1}^{n}r\left(s_{t},a_{t}\right)\right]\\ \; & =E_{\pi_{\theta }}\left[A^{\pi_{\theta }}\left(s_{t},a_{t}\right)\nabla_{\theta }\log\pi_{\theta }\left(a_{t}^{\left(n\right)}|s_{t}^{\left(n\right)}\right)\right] \end{array}$$
where $A^{\pi_{\theta }}$ means the advantage function. It represents the advantage of action $a_{t}$ relative to other actions in state $s_{t}$, which is equivalent to the deviation of the variable relative to the mean value.

Finally, θ is updated by gradient ascent, and the optimal parameter $\theta^{*}$ is obtained by multiple iterations.
(7)$$\theta \leftarrow \theta +\alpha \nabla_{\theta }J\left(\theta \right)$$

The critic network evaluates the actions selected by the actor network and fits the cumulative return of each state-action pair. The loss function can be expressed as
(8)$$L\left(\theta \right)=\frac{1}{2}\sum_{s_{t},a_{t}}{\left[Q\left(s_{t},a_{t}\right)-Q_{\theta }\left(s_{t},a_{t}\right)\right]}^{2}$$
where *Q* and $Q_{\theta }$ are, respectively, the actual action value and the fitted action value estimated by the network. The optimization goal of the algorithm is to minimize the gap between the two, so the update of the network parameter θ is:(9)$$\theta \leftarrow \theta +\alpha \left[Q\left(s_{t},a_{t}\right)-Q_{\theta }\left(s_{t},a_{t}\right)\right]\nabla Q\left(s_{t},a_{t}\right)$$
(10)$$\theta \leftarrow \theta +\alpha \left[r\left(s_{t},a_{t}\right)+Q_{\theta }\left(s_{t+1},a_{t+1}\right)-Q_{\theta }\left(s_{t},a_{t}\right)\right]\nabla Q\left(s_{t},a_{t}\right)$$

The improvement of the SAC algorithm based on the AC framework is mainly to maximize the entropy of the action while maximizing the cumulative reward, which can make the selected action more random at each step, and thus increases the exploration ability of the agent and avoids the centralized selection of one action. The agent trained by the SAC algorithm has better exploration and robustness, and can avoid premature convergence to a local optimum. The entropy of the action is expressed as:(11)$$H_{\pi \left(a_{t+1}|s_{t+1}\right)}=-E_{a}\left[\log\pi \left(a_{t+1}|s_{t+1}\right)\right]$$

The entropy of the action is considered while calculating the cumulative reward, so the optimization goal is changed to:(12)$$\begin{array}{c} J\left(\theta \right)=\frac{1}{2}E_{\left(s_{t},a_{t}\right)}{\left[r\left(s_{t},a_{t}\right)+\gamma \left(Q_{\theta }\left(s_{t+1},a_{t+1}\right)+\alpha H_{\pi \left(a_{t+1}|s_{t+1}\right)}\right)-Q_{\theta }\left(s_{t},a_{t}\right)\right]}^{2} \end{array}$$
where α is the temperature coefficient of the SAC algorithm, which controls the randomness of the policy π. Usually, α is adaptively changed during training to adjust the weight of entropy $H_{\pi }$ in the overall optimization goal.

This paper proposes the R-SAC algorithm, which modifies the network structure of SAC, adding a residual-like structure. The residual-like structure adds skip connections to each layer of the SAC network; in other words, it adds the state input to the output of each layer as a joint input to the next layer of the network. Appropriately increasing the layers of the neural network can enhance the network’s expressive ability and improve performance to a certain extent, but at the same time, it may cause problems such as gradient vanishing. After adding the residual-like network structure, since the derivative contains an identity entry, effective back-propagation can be maintained. On the other hand, the residual-like network structure can supplement the feature information of state input lost through the network, which is more conducive to the autonomous learning of the agent.

Our residual-like network refers to the residual block structure of ResNet [41]. As shown in Figure 2, the principle is directly introducing a short connection from the input to the output of the nonlinear layer, so the mapping becomes
(13)$$y=f\left(x,w\right)+x$$
where *x* denotes the input and *w* is the network parameters.

In short, residual structure is just a way of building the network. The difference between our residual-like network with the standard residual block is that our skip connections are not connected every two layers as commonly used, but every layer performs this operation. The network structure is shown in Figure 3.

Next we discuss why the residual structure can alleviate the training problem of deep networks. Suppose the mapping of the residual block of the ith layer is
(14)$$F\left(x_{i},w_{i}\right)=x_{i}+f\left(x_{i},w_{i}\right)$$

Then the input of the *i* + 1th layer is
(15)$$\begin{array}{cc} x_{i+1} & =F\left(x_{i},w_{i}\right)\\ \; & =x_{i}+f\left(x_{i},w_{i}\right) \end{array}$$

Loop this calculation, we get the input of the *N*th layer
(16)$$x_{N}=x_{i}+\sum_{n=i}^{N-1}F\left(x_{n},w_{n}\right)$$

The above formula shows good back-propagation characteristics. Assuming that the loss is *l*, according to the chain derivation rule, we can get
(17)$$\begin{array}{cc} \frac{\partial l}{\partial x_{i}} & =\frac{\partial l}{\partial x_{N}}\frac{\partial x_{N}}{\partial x_{i}}\\ \; & =\frac{\partial l}{\partial x_{N}}\left(1+\frac{\partial }{\partial x_{i}}\sum_{n=i}^{N-1}F\left(x_{n},w_{n}\right)\right) \end{array}$$

From the formula, it can be seen that the gradient consists of two parts, which are, respectively, the value weighted by networks and the value without weighting. The linear connection of these two parts ensures that information can be back-propagated to shallow layers, and the gradient will not be 0 even if the weights are very small. Therefore, the vanishing gradient problem does not exist in a residual-like network.

### 3.2. Offline Expert Experience

In this paper, the training of reinforcement learning is assisted by offline expert experience supervised learning. In the implementation of AlphaGo in 2015 [42], a large number of human chess manuals are used to pre-train and initialize the algorithm networks before the algorithm training. After that, AlphaGo learns more unknown chess compositions through self-play. It can be seen that supervised learning by expert experience can raise the lower limit of the agent and greatly reduce the time of training. After pre-training by expert experience, reinforcement learning is an indispensable method to increase the upper limit of the agent. In fact, expert experience has certain limitations, and more unknown situations can only be met by continuous exploration of the agent. In the process of interacting with the environment, it learns to decide the optimal action in different states, and it also has a certain adaptability of the unknown state.

The replay buffer in conventional reinforcement learning can only replay the experience for one training. In order to improve the training efficiency, this paper collects a sufficient amount of expert experience for storage before training, and directly reads the offline expert experience and conducts supervised learning first. Then the R-SAC algorithm is trained after the pre-trained model is finished. Since the random exploration in the early stage is saved in various scenarios, the training efficiency is greatly improved, and the training mode of one exploration and multiple utilization is realized. The flow charts of expert experience pre-training and the whole training process are shown in Figure 4 and Figure 5.

### 3.3. Multi-Step Td-Error

In terms of reward function, because of the advantages of the R-SAC algorithm, the reward function can be greatly simplified. We judge whether the agent is closer to the target point than the previous step, and decide to give it a positive reward or a negative reward. If it collides with the obstacles on the way, a larger negative reward will be given. If it moves beyond the bound of the map, a specific negative reward will be given. Finally, a maximum positive reward will be given if it reaches the final target point. This mode may have a drawback, as shown in Figure 6.

If the agent misses the target point, it will try to turn and correct to the target point. During this process, it may move away for a short time. However, at this time it will get the same negative reward, which makes the agent be confused and not know what to do and where to go. In fact, as long as this short-term deceleration process is finished, the subsequent reverse acceleration process can make it closer to the target point and trigger a positive reward. However, the current reward mode prevents it from seeing future positive rewards, and only gets stuck with immediate negative rewards. Therefore, we optimize the original single-step TD error and change it to multi-step TD error to calculate rewards, so that it can comprehensively consider the subsequent multi-step rewards. The update mechanism of multi-step TD error is
(18)$$G^{\left(n\right)}=r^{\left(t+1\right)}+\gamma r^{\left(t+2\right)}+\cdots +\gamma^{n-1}r^{\left(t+n\right)}+\gamma^{n}Q^{\left(t+n+1\right)}\left(s_{t+n},a_{t+n}\right)$$

Therefore, the optimization goal of multi-step R-SAC algorithm is:(19)$$J\left(\theta \right)=\frac{1}{2}E_{\left(s_{t},a_{t}\right)}{\left[r^{\left(n\right)}\left(s_{t},a_{t}\right)+\gamma^{\left(n\right)}\left(Q_{\theta }\left(s_{t+n+1},a_{t+1}\right)+\alpha H_{\pi \left(a_{t+1}|s_{t+1}\right)}\right)-Q_{\theta }\left(s_{t},a_{t}\right)\right]}^{2}$$

**Figure 5 sensors-22-04316-f005:**
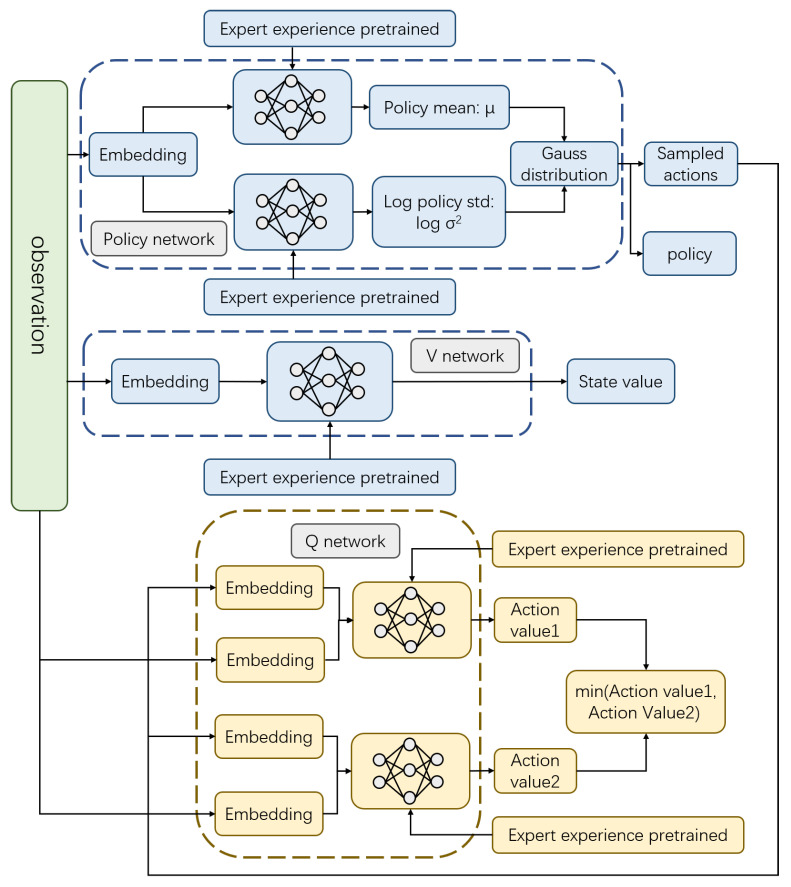
The training process of R-SAC algorithm.

**Figure 6 sensors-22-04316-f006:**
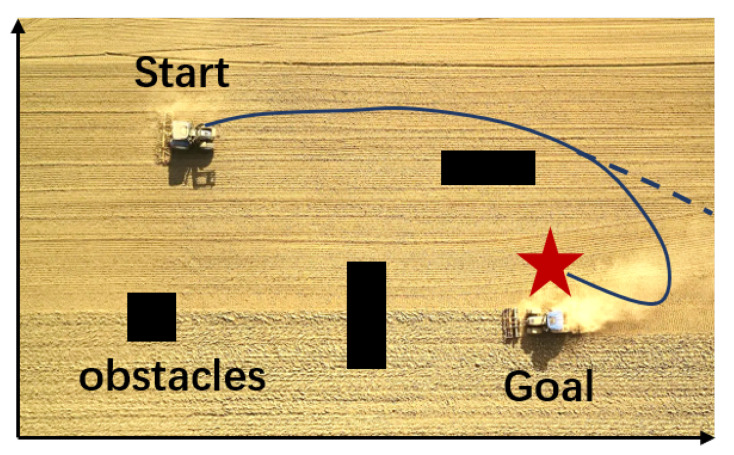
The drawback existed in the training process. If the agent misses the goal point, it is difficult to adjust the action to turn to the correct direction.

## 4. Experiments and Results

We take agricultural robots as the research object and carry out groups of path planning experiments. When the robot executes agricultural tasks, it needs to control its direction automatically, which is essentially a path planning task. In the farmland scenario, there are always some obstacles, such as working people and other agricultural machinery. Therefore, the robot must have the ability of autonomous obstacle avoidance. We establish a 600 × 600 two-dimensional environment, and set eight rectangular obstacles in the environment, as shown in Figure 7; the black rectangles represent the obstacles, and the red pentagram represents the goal point. The goal of the robot is to avoid these obstacles and reach the target point safely. Meanwhile, the optimal path is as short as possible, and the consumed time is as fast as possible. The experiments are based on Python 3.7 and performed on Ubuntu 18.04 with NVIDIA RTX 3070 GPU and 16G RAM. Considering that reinforcement learning makes decisions through the interaction of each step, the experiment evaluation metrics include the number of decision steps and the length of the entire path. The main parameters of the algorithm and the location and shape settings of obstacles are shown in Table 1 and Table 2. After the algorithm is trained, we save the trained model for testing in the same environment. During the test, we record the position of the agent and plot it, and calculate the steps and path lengths it takes.

### 4.1. Static Obstacles Environment

First of all, we conduct different target points experiments with static obstacles based on our proposed algorithm. As shown in Figure 7, the start point is (0, 0), and the target point is set at (500, 500). There are obstacles in the map, so the agent must learn to avoid collision and detour to reach the target point. The path in the figure verifies the collision avoidance ability of the agent. The agent has indeed bypassed several obstacles in the forward direction and reached the destination in a roundabout way. Moreover, the agent chooses a closer path, which is the result that comprehensively considers efficiency and safety. Figure 8 shows the paths with another two target points, respectively set at (200, 400) and (400, 400), and our R-SAC algorithm performs well in these cases. Faced with different destinations, the agent can independently decide to pass through different obstacles and choose different paths. They successfully complete the task under the condition of saving steps as much as possible.

**Figure 7 sensors-22-04316-f007:**
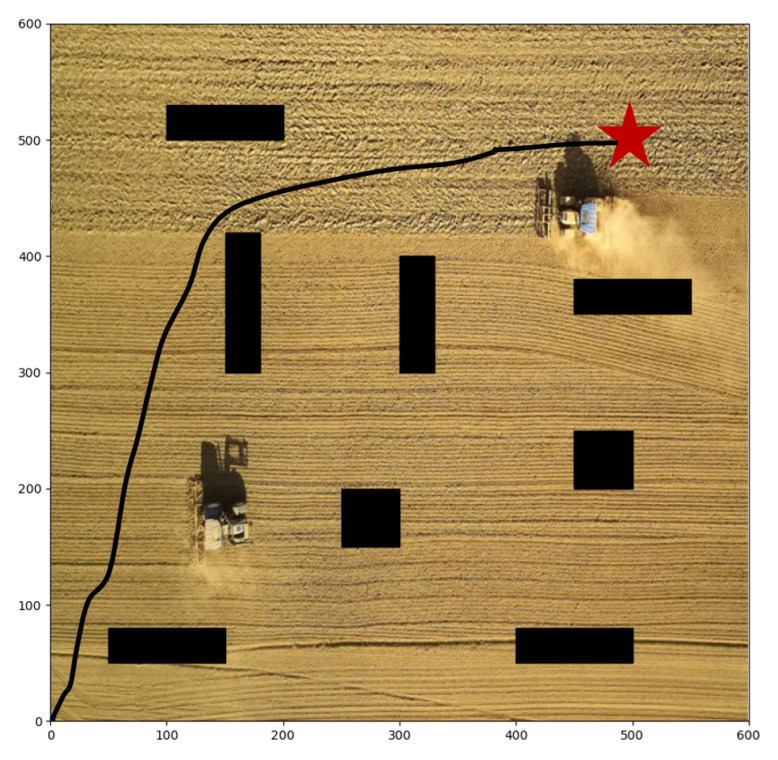
The planned path with start point of (0, 0) and target point of (500, 500).

Table 3 shows the experiment results of other target points that we supplement based on the same algorithm. We select target points in multiple directions to comprehensively test the performance of the R-SAC algorithm. The results show that our method has stable performance under conditions of different directions, different obstacle avoidance difficulties, and different distances. The robot completes the path planning task in shorter steps and path lengths. Figure 9 and Table 4 show the experiment results with the target point of (500, 500) compared with the commonly used reinforcement learning algorithms DQN and Dueling Double DQN (D3QN). Obviously, the DQN algorithm does not perform well. Although its reward has a tendency to converge, the convergence speed is too slow, so it cannot converge to the maximum reward value within 20,000 episodes. From the results in Table 4, it can be seen that DQN finds a feasible path, but it is not optimal, and the steps and path length of DQN algorithm are much higher than the other two. D3QN algorithm achieves better performance because it utilizes double networks to estimate the Q value, and thus solves the overestimation problem of DQN. Moreover, it uses the dueling architecture to separate the state from the action so that it learns more accurately. D3QN algorithm converges faster, but some fluctuations still exist after convergence, and the steps and path lengths it consumed are also larger than those of the R-SAC algorithm. Our R-SAC algorithm comprehensively considers the Q value and action entropy, which avoids the local optimum problem. At the same time, the residual-like network enables it to fully learn the features of the input state, so the obtained path is the shortest. Figure 9 denotes that R-SAC network has the fastest convergence speed and remains stable after convergence.

### 4.2. Dynamic Obstacle Environment

In actual agricultural scenarios, obstacles in the farmland, such as agricultural machinery and working people, are always constantly moving, so robots are required to have a certain ability to adapt to dynamic obstacles. Based on the obstacles above-mentioned, we assume that all obstacles move toward a random direction at each step, which increases the difficulty of obstacle avoidance. Figure 10 shows the path under the condition that the target point is (500, 500), and the position of the obstacles shown in the figure is at the last moment. Compared with the static situation of Figure 7, the path in Figure 10 passes through the middle obstacles instead of traversing from the outer obstacles. The reason for this may be that the obstacles tend to move outward during the training process, so they free up the middle area, giving the agent room to move through the middle. Figure 11 shows the results with the target points (200, 400) and (400, 400). Compared with the static experiments in Figure 8, the paths are also significantly different. The path finally earned by the agent is under the influence of dynamic obstacles, so it has a certain ability to adapt to the movement of obstacles.

Table 5 shows the dynamic obstacle avoidance results of other target points. Compared with Table 1, the path changes to become either longer or shorter. Since the obstacles move randomly at each step, the position of the obstacles faced by the agent during the training process is different, and the optimal path learned in the end is also different. Figure 12 and Table 6 show the comparison results of the algorithms under the condition of (500, 500) target points. Contrast with the static results, three algorithms are affected by dynamic obstacles, the optimal paths and training curves change as well. The steps consumed by the R-SAC algorithm is greatly reduced. Corresponding to Table 4, it can be seen that the steps are saved by passing through the middle obstacles, and the path length is correspondingly shortened. From the reward curve, R-SAC algorithm converges as fast as before and has high stability. The D3QN algorithm maintains the result similar to that of the static experiments, probably because the obstacle movement does not have much influence on the path of the agent. The over-estimation problem of DQN makes its convergence relatively difficult, and the training process is very unstable. It can be seen from the table that the consumed steps are significantly increased compared to the static scenario, which indicates that the eventual path is probably not the optimal path. In summary, our proposed R-SAC algorithm has stable performance in static and dynamic obstacles, and is better than other algorithms, which verifies the effectiveness of the algorithm.

## 5. Discussions

### 5.1. Multi-Step Comparison

In order to compare the influence of the Multi-step mechanism on the algorithm convergence, we conduct the contrast experiment with the single-step, 10-step and 5-step mechanism, and other configurations are the same as the algorithm proposed in this paper. From the point of view of the reward curve and TD-error curve in Figure 13 and Figure 14, the 5-step mechanism has a certain performance improvement compared to the single-step. Due to the combination of reward evaluation in the next five steps, the agent has long-term planning and makes more reasonable decisions, so it has a more stable training curve. However, when we increase the multi-step to 10, the performance becomes worse. It is speculated that the agent considers too many steps, which increases the learning difficulty of the agent. It may take too many factors into consideration for a simple action and ignore the most important reward at the current step, thus falling into confusion. Therefore, there is a certain range for the choice of multi-step. In general, the long-term consideration of three-five steps is more conducive to the training of the agent.

### 5.2. Ablation Experiments

For the sake of verifying the advantages of several improved mechanisms we proposed, we conduct ablation experiments on residual-like network, offline expert experience, and multi-step TD-error mechanisms. Figure 15 and Figure 16 show the trends of rewards and TD-error during training. In the figure, the scheme “w/o Expert Experience” means we only use the R-SAC algorithm with multi-step TD-error. The scheme “w/o Residual-like Network” means we only use SAC algorithm with offline expert experience and multi-step TD-error mechanism without residual-like structure. The scheme “w/o Multi-step TD-error” means we only use the R-SAC algorithm with offline expert experience. The scheme “Full” means our proposed complete algorithm.

From the results in Figure 15 and Figure 16, the offline expert experience mechanism has the greatest impact on the algorithm. Directly using reinforcement learning for training without expert experience is too hard for the agent to learn the task of this paper, and it is difficult to quickly explore the correct direction, so the convergence speed is slow. Collecting expert experience in advance for supervised learning as the initialization of reinforcement learning network solves this problem. The initial network of the agent has a good performance, and the later training only needs to adjust on this basis, which is equivalent to reducing the difficulty of the task. Secondly, the residual-like network has a certain influence. If the residual-like network is removed and the fully connected layer is used instead, the input state information may be lost when passing through the networks, thus affecting the network performance. In addition, we remove the multi-step TD-error mechanism and directly use the single-step error calculation. As a result, the agent may not see the long-term feedback after a few steps, resulting in a “short-sighted” problem, which is also the reason for the instability of the green curve in the later stage. Finally, the complete algorithm with all the improved mechanisms converges fastest and remains at a stable level to the end.

Table 7 compares the difference of steps and the path length in the ablation experiments. The R-SAC algorithm without expert experience is relatively the longest path. Due to its slow convergence speed, it may not converge to the shortest path in 20,000 episodes. The algorithm without residual-like network and without multi-step TD-error mechanism lead to a few steps difference from the complete algorithm. Although the gap is not large, the number of steps is an important evaluation index for path planning problems. The gap of a few steps determines whether the path is optimal, and reflects the performance gap of the algorithm. Therefore, our proposed R-SAC algorithm with all improved mechanisms is an efficient, safe, and stable path planning method.

## 6. Conclusions

This paper focuses on the application of reinforcement learning algorithms in agricultural scenarios, and completes the tasks of obstacle avoidance and path planning for agricultural robots. In general, this paper proposes a novel R-SAC algorithm, which solves the vanishing gradient problem and prevent the algorithm losing some state information during the training process. This innovation improves the performance of the algorithm, and enables the robot to have a stable performance with various target points in static and dynamic scenarios. Secondly, this paper proposes a method that utilizes offline expert experience for supervised learning, and uses the pre-training results as the initialization parameters of the reinforcement learning network, which greatly improves the training efficiency of the algorithm. In addition, this paper adds a multi-step TD-error mechanism to the algorithm update, so that the robot has the ability to comprehensively consider long-term feedback and adjust the policy flexibly. It is proved that our proposed method is more stable and efficient by contrasting experiments with other algorithms. We also conduct ablation experiments to analyze the various improved modules of the algorithm to verify the effectiveness of these improved mechanisms. In future research, we will further combine the real scenarios and robot control knowledge to make it closer to the real applications, so that the algorithm can be implemented in physical experiments.

## Figures and Tables

**Figure 1 sensors-22-04316-f001:**
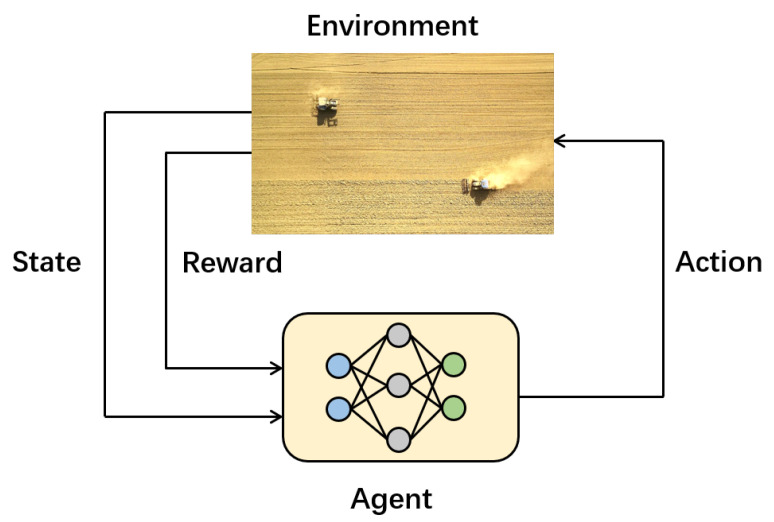
The framework of reinforcement learning.

**Figure 2 sensors-22-04316-f002:**
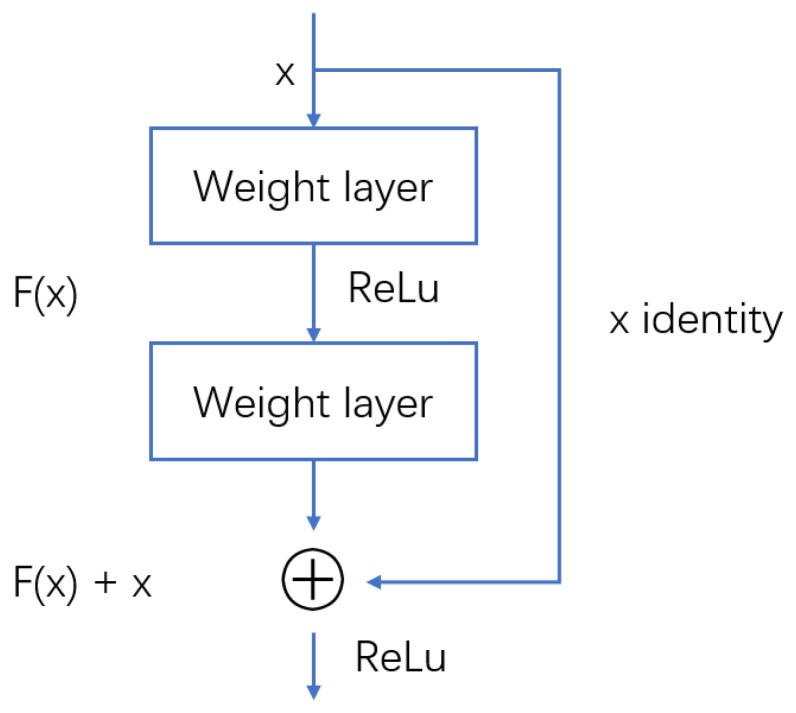
The structure of residual block in ResNet.

**Figure 3 sensors-22-04316-f003:**
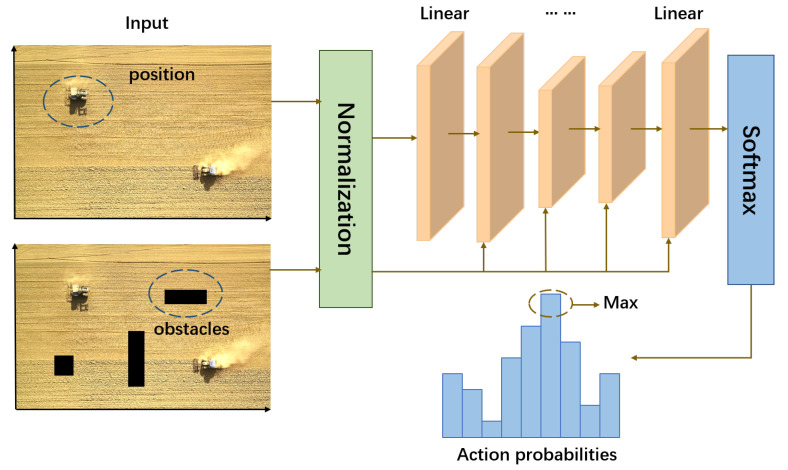
The residual-like network used in reinforcement learning.

**Figure 4 sensors-22-04316-f004:**
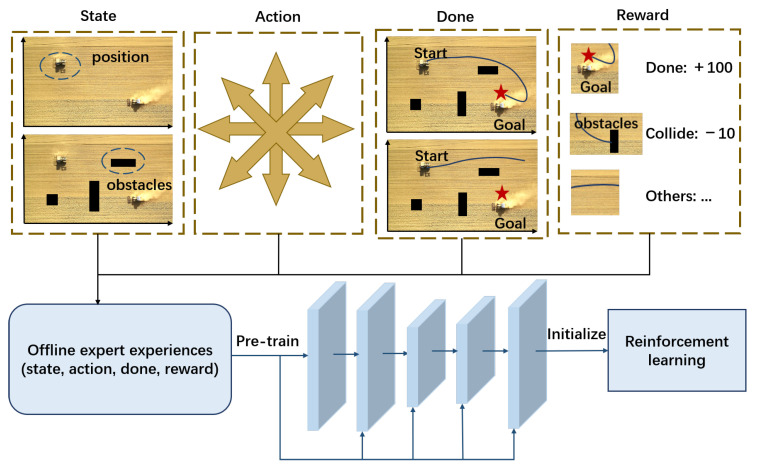
The offline expert experience pre-training mechanism.

**Figure 8 sensors-22-04316-f008:**
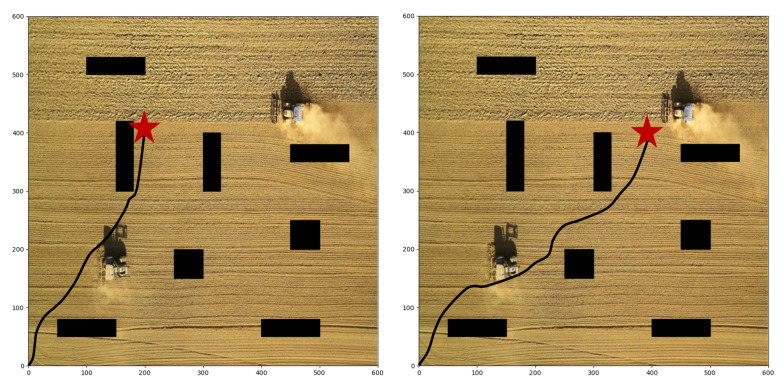
The planned paths with target point of (200, 400) and (400, 400).

**Figure 9 sensors-22-04316-f009:**
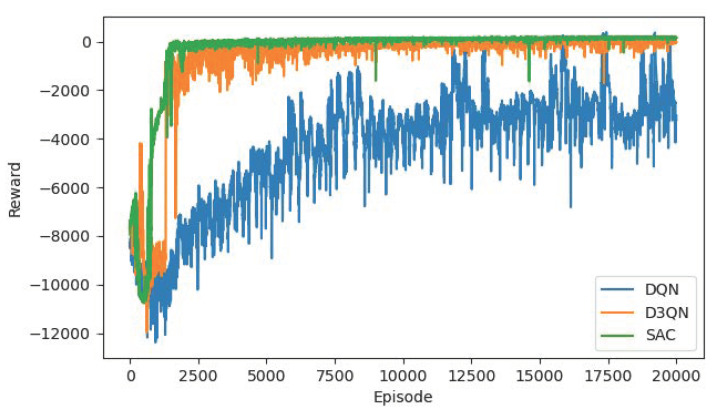
The reward curves of DQN, D3QN, R-SAC algorithm with target point of (500, 500).

**Figure 10 sensors-22-04316-f010:**
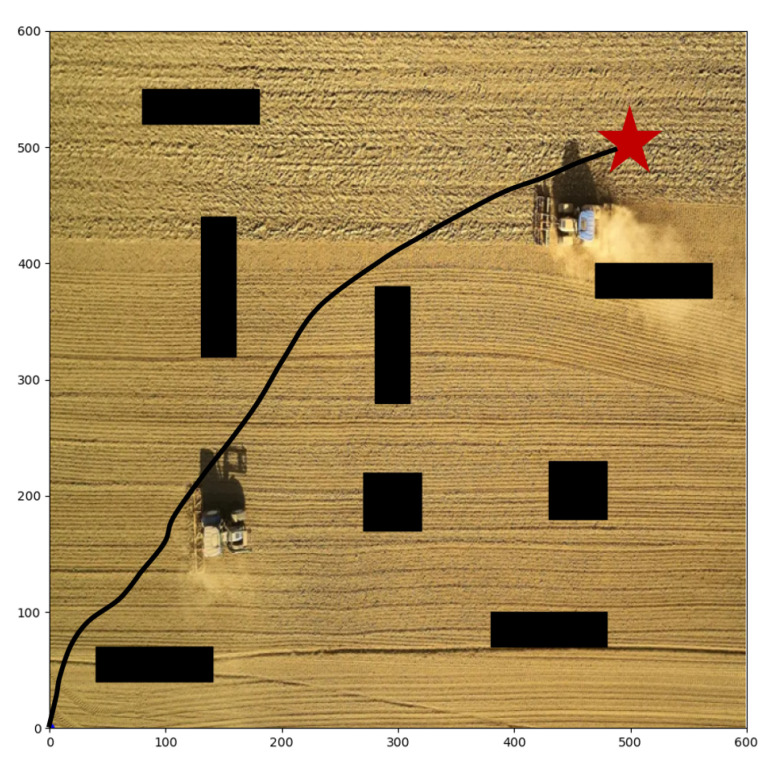
The planned path with target point of (500, 500) and dynamic obstacles.

**Figure 11 sensors-22-04316-f011:**
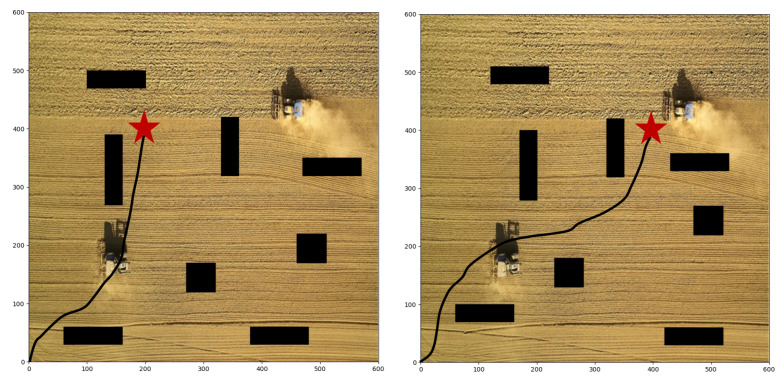
The planned paths with target point of (200, 400) and (400, 400) and dynamic obstacles.

**Figure 12 sensors-22-04316-f012:**
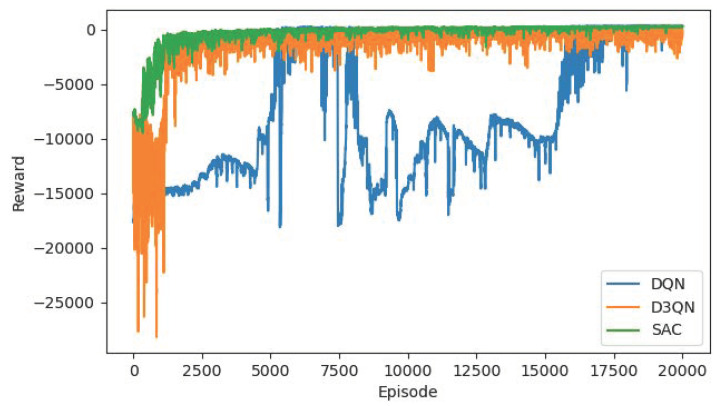
The reward curves of DQN, D3QN, R-SAC algorithm with target point of (500, 500) and dynamic obstacles.

**Figure 13 sensors-22-04316-f013:**
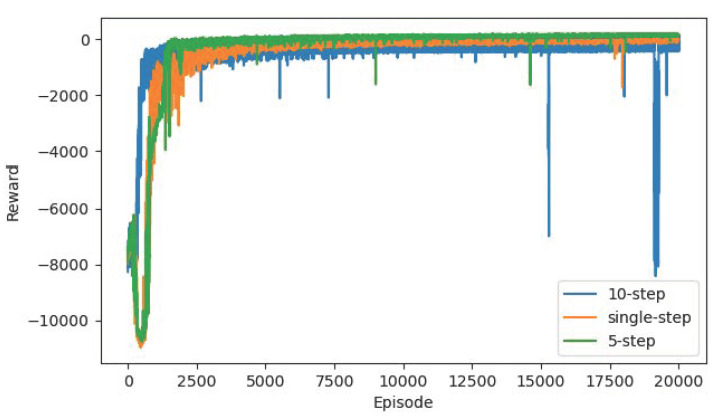
The reward curves of single-step, 10-step and 5-step mechanism based on R-SAC algorithm.

**Figure 14 sensors-22-04316-f014:**
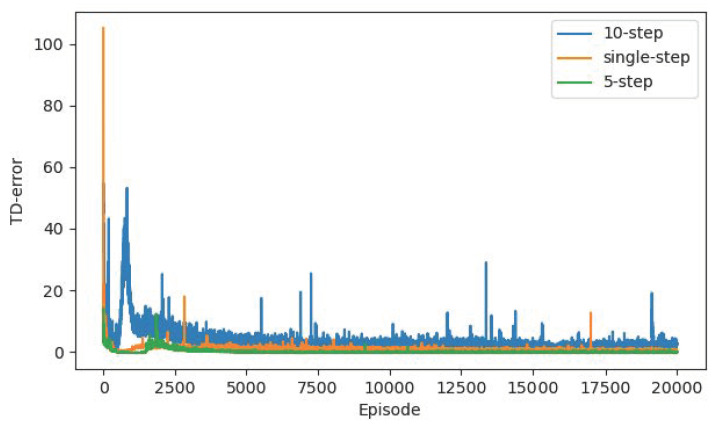
The TD-error curves of single-step, 10-step and 5-step mechanism based on R-SAC algorithm.

**Figure 15 sensors-22-04316-f015:**
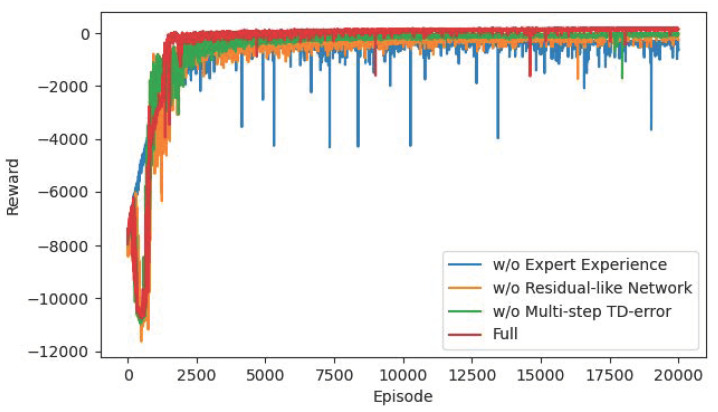
The reward curves in the ablation experiment.

**Figure 16 sensors-22-04316-f016:**
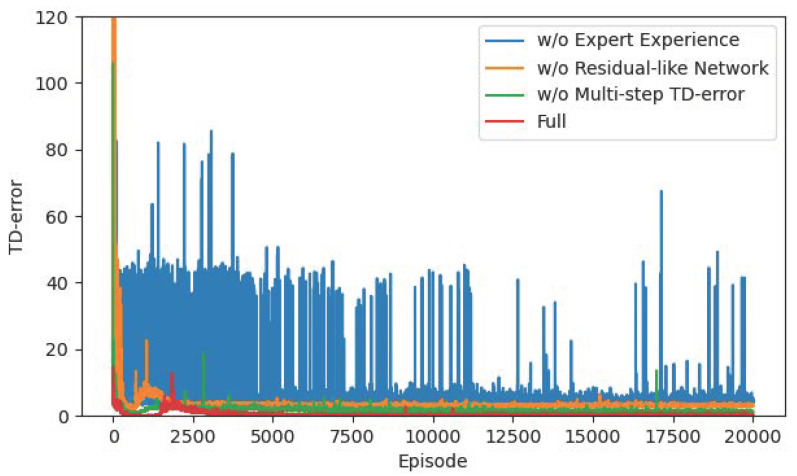
The TD-error curves in the ablation experiment.

**Table 1 sensors-22-04316-t001:** The main parameters of the algorithm.

Parameter	Value
Batch size	$2^{7}$
Buffer size	$2\times 10^{4}$
Maximum training episodes	$2\times 10^{4}$
Maximum time steps	$10^{3}$
Discount factor	0.99
Network hidden size	$2^{8}$
Learning rate	$5\times 10^{-4}$

**Table 2 sensors-22-04316-t002:** The position of the lower left vertex, width, and height of the rectangular obstacles.

Position	Width	Height
(50, 50)	100	30
(150, 300)	30	120
(100, 500)	100	30
(450, 350)	100	30
(250, 150)	50	50
(400, 50)	100	30
(300, 300)	30	100
(450, 200)	50	50

**Table 3 sensors-22-04316-t003:** The steps and path length of R-SAC algorithm with different target points.

Target Point	Steps	Path Length
(500, 500)	171	860.87
(600, 600)	198	996.94
(400, 400)	142	715.78
(600, 100)	149	754.39
(200, 400)	128	643.94
(300, 500)	148	748.44
(600, 300)	167	842.05

**Table 4 sensors-22-04316-t004:** The steps and path length of DQN, D3QN, R-SAC algorithm with target point of (500, 500).

Algorithm	Steps	Path Length
DQN	195	981.34
D3QN	176	887.28
R-SAC	171	860.87

**Table 5 sensors-22-04316-t005:** The steps and path length of R-SAC algorithm with different target points and dynamic obstacles.

Target Point	Steps	Path Length
(500, 500)	159	800.17
(600, 600)	192	967.26
(400, 400)	145	732.95
(600, 100)	155	780.65
(200, 400)	132	665.36
(300, 500)	150	758.34
(600, 300)	172	873.45

**Table 6 sensors-22-04316-t006:** The steps and path length of DQN, D3QN, R-SAC algorithm with target point of (500, 500) and dynamic obstacles.

Algorithm	Steps	Path Length
DQN	218	1097.56
D3QN	172	866.15
R-SAC	159	800.17

**Table 7 sensors-22-04316-t007:** The steps and path length in the ablation experiment.

Algorithm	Steps	Path Length
w/o Expert Experience	192	967.56
w/o Residual-like Network	177	893.81
w/o Multi-step TD-error	174	876.42
Full	171	860.87

## Data Availability

Not applicable.

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
