# Peer review of "The Intelligent Path Planning System of Agricultural Robot via Reinforcement Learning"

_sensors, 2022, doi:10.3390/s22124316_

Round 1

Reviewer 1 Report

ID: sensors-1740403-peer-review-v1

Title: The Intelligent Path Planning System of Agricultural Robot via Reinforcement Learning

The focus here is on a Residual-like Soft Actor Critic (R-SAC) algorithm for agricultural scenarios to realize safe obstacle avoidance and intelligent path planning of robots. Furthermore, authors proposed reinforcement learning (RL) methods to train agricultural robots to plan paths autonomously.

In general, the paper does not have a good quality. The topic may be interest of robotics and computer science researchers. I have some comments about organization and content of the paper. Please see the following comments:

- My main concern is about the methodology of the paper. I do not see much about RL to be considered as solution methods studied deeply. I know that RL is not preferable for solving simple problems, and also it requires a lot of data and a lot of computation. So, too much reinforcement learning can lead to an overload of states which can diminish the results. How could authors overcome this difficulty?

- Another concern is that the language in the paper is sometimes too informal for academic writing. For example, I would avoid contractions in formal writing and "what’s more" (in Page 1) might come across as conversational. I have not seen "What is more" in many academic papers, but "moreover" and "furthermore" are pretty well used.

- The literature review section lacks discussion and summary. Ideally, it should locate before Section 3 of the current revision. This should make reader ready for Section 3 considering gaps resulted from Section 2 (Why RL has obvious advantages for the use of agricultural robots in dynamic environments?).

- Never use etc. at the end of a series that begins with for example, e.g., such as, and the like, because these terms make etc. redundant: they already imply that the writer could offer other examples. 

example:

Page 9: such as working people, other agricultural machinery, etc.

- The title of most papers in the reference list are lowercase, whereas few are Capitalized each Word. This is a case of inconsistency. Please make all of them lowercase for the sake of simplicity.

- Authors should avoid abbreviations in the keyword list:

example:

IoT applications --> Internet of Things

- Scope of work: As the origin of collision/deadlock avoidance (not a new concept), it has been widely studied in the manufacturing domain for decades.  At the beginning of Section 1, my suggestion is that authors briefly refer to the origin with citing a couple of papers. [a] deadlock-free scheduling of manufacturing systems using petri nets and dynamic programming, IFAC, vol.32, pp. 4870-4875 [b] resolution of deadlocks in a robotic cell scheduling problem with post-process inspection system: avoidance and recovery scenarios, 2015 IEEE IEEM, 2015, pp. 1107-1111

- The use of English is not professional. Please see the following errors:

Page 3: Santos*† --> Santos

Page 8: as shown in the Figure 5 --> as shown in Figure 5

...

Reviewer 2 Report

This paper presents an approach for path planning and obstacle avoidance using reinforcement learning. It aims to be adopted for an agricultural robot navigating in the field. The clear motivation and background are provided, the description of the method is presented with some experimental results. However, a few concerns need to be addressed. First, the major contribution might be the R-SAC algorithm, an offline expert experience pre-training, and multi-step TD-error adopted. In Section 3, especial Section 3.1, most materials are given by the existing SAC algorithm. There are relatively less contents related to the proposed Residual-like SAC. The authors only use three figures for illustration, but without detailed explanation. Second, the literature survey and reference are not comprehensive. To address the path planning and obstacle avoidance for real world applications, it generally requires some initial trajectory planning prior to the operation. Please refer to

Lin, H.Y. and Peng, X.Z., 2021. Autonomous Quadrotor Navigation With Vision Based Obstacle Avoidance and Path Planning. IEEE Access, 9, pp.102450-102459.
Han, G., Zhou, Z., Zhang, T., Wang, H., Liu, L., Peng, Y. and Guizani, M., 2020. Ant-colony-based complete-coverage path-planning algorithm for underwater gliders in ocean areas with thermoclines. IEEE Transactions on Vehicular Technology, 69(8), pp.8959-8971.

Third, the authors propose to use multi-step TD error instead of single-step. It is necessary to provide the drawbacks for multi-step TD error. It is also expected to present the simulation results for these two metrics. Fourth, there are only several results presented with Figures 6, 7, 9, 10, along with a few tables. These do not explicitly show how the proposed algorithm is carried out for the results. The experiments mainly present the simulation for path planning from a starting point to some target points, it is hard to judge the obstacle avoidance is performed. Fifth, this work basically provides an approach for path planning in the simulation environment. In the experiments, It is not agriculture related. The authors provide a map with several obstacle blocks for path planning. If this paper claims to be used for agricultural applications, there should be some real world scenarios directly from the field. Finally, there are many grammar mistakes which should be corrected.

Round 2

Reviewer 1 Report

I have read the paper once more, mostly with focus on my comments. 

The paper has a better presentation in the current format. Also, it scientifically sounds now. 

The paper can be accepted as it is.

Reviewer 2 Report

The authors have addressed most of the reviewer's concerns in this revision.